# Virus-like Particles Carrying a Porcine Circovirus Type 2b Peptide Induce an Antibody Response and Reduce Viral Load in Immunized Pigs

**DOI:** 10.3390/vaccines14010024

**Published:** 2025-12-24

**Authors:** Ana del Socorro Hernández-Aviña, Marco Antonio Cuéllar-Galván, Jorge Alberto Salazar-González, Oscar Alejandro Albarrán-Velázquez, María de los Ángeles Beltrán-Juárez, René Segura-Velázquez, Sara Elisa Herrera-Rodríguez, Abel Gutiérrez-Ortega, José Iván Sánchez-Betancourt

**Affiliations:** 1Unidad de Biotecnología Médica y Farmacéutica, Centro de Investigación y Asistencia en Tecnología y Diseño del Estado de Jalisco, Normalistas 800, Colinas de la Normal, Guadalajara 44270, Jalisco, Mexico; anhernandez_al@ciatej.edu.mx (A.d.S.H.-A.); jasalazar@ciatej.mx (J.A.S.-G.); 2Unidad de Investigación, Facultad de Medicina Veterinaria y Zootecnia, Universidad Nacional Autónoma de Mexico, Universidad 3000, Copilco, Coyoacán, Ciudad de México 04510, Mexico; macgalvan0205@gmail.com (M.A.C.-G.); alexhale911@gmail.com (O.A.A.-V.); rsegura@fmvz.unam.mx (R.S.-V.); 3Departamento de Investigación y Desarrollo, Sanfer Salud Animal, 7 Norte 416, Centro, Tehuacán 75700, Puebla, Mexico; mabeltran@corex.com.mx; 4Unidad Sureste, Centro de Investigación y Asistencia en Tecnología y Diseño del Estado de Jalisco, km 5.5 Carretera Sierra Papacal-Chuburná, Chuburná, Mérida 97302, Yucatán, Mexico; sherrera@ciatej.mx; 5Departamento de Medicina y Zootecnia de Cerdos, Facultad de Medicina Veterinaria y Zootecnia, Universidad Nacional Autónoma de Mexico, Universidad 3000, Copilco, Coyoacán, Ciudad de México 04510, Mexico

**Keywords:** porcine circovirus type 2, subunit vaccine, chimeric virus-like particles, Cap peptide

## Abstract

**Background**: Porcine circovirus type 2 (PCV2) remains one of the most important pathogens that infects swine, causing considerable economic losses worldwide. PCV2 vaccines are commercially available, and the development of experimental vaccines that could confer better protection against emerging genotypes is underway. The expression of virus-like particles (VLPs) carrying different PCV2 capsid (Cap) peptides in *E. coli* was recently reported. These chimeric particles were adjuvated with an oil-in-water emulsion with polymer and induced different titers of serum IgG in BALB/c mice after a single subcutaneous injection. The aim of this study was to assess the immune response and protective efficacy elicited by VLPs carrying the PCV2b Cap carboxy-terminal peptide in the target species. **Methods**: Domestic pigs (*Sus scrofa domesticus*) were immunized intramuscularly with 25 μg of adjuvated chimeric VLPs on days 0 and 14 and challenged on day 28 with a PCV2b Mexican isolate. PCV2 peptide-specific IgG seroconversion, serum cytokines, viral load in nasal swabs and organs, and histopathological score were determined. **Results**: IgG levels peaked 28 days post-immunization. Interleukin-12 and -18 and interferon-gamma increased 21 days after immunization. In addition, genomic material of PCV2 was detected in nasal swabs from one specimen on day 7, two specimens on day 14, and two specimens on day 21 following viral challenge. Finally, histological lesions were not less severe in immunized specimens compared to non-vaccinated/challenged specimens. **Conclusions**: These results suggest that immunization with chimeric VLPs could contribute to controlling viral shedding in pig herds where a PCV2b genotype is most prevalent.

## 1. Introduction

Porcine circovirus type 2 (PCV2) is acknowledged as a widely distributed pathogen that has a significant impact on the global swine industry [1]. This virus is the primary causative agent of porcine circovirus-associated disease (PCVAD), which commonly appears as post-weaning multisystemic wasting syndrome, reproductive failure, enteritis, or respiratory disease, having substantial detrimental effects on swine health and production [2,3].

PCV2 is a small non-enveloped virus with a 17–20-nanometer icosahedral structure and a circular, single-stranded DNA genome of 1.7 kb that belongs to the Circoviridae family [4]. Eleven open reading frames (ORFs) have been identified [5], and ORF2 encodes for the capsid protein (Cap), which is the main antigenic determinant of PCV2 and is used for PCV2 genetic diversity assessment [6,7]. To date, eight PCV2 genotypes have been reported (PCV2a to PCV2h). PCV2a was the first genotype that appeared in farms from 1996 to 2000. Then, numerous publications have discussed a transition from PCV2a to PCV2b, which may be associated with the appearance of more severe clinical signs in North America and Europe [8,9]. PCV2c was initially identified in Denmark, and a second genotype shift from PCV2b to PCV2d seems to be happening worldwide [10], highlighting the elevated mutation rate that this virus undergoes [11].

Commercial PCV2 vaccines have been developed to mitigate or prevent PCVAD in pigs. These vaccines contain the inactivated PCV2a virus [12,13], chimeric PCV1-2a virus [3,14], or PCV2a Cap virus-like particles (VLPs) [15,16]. Experimental PCV2 vaccines have also been reported, like attenuated chimeric viruses [17], genetic vaccines [18], virus-like particle subunits [19,20,21,22], and viral vectored vaccines [23,24]. PCV2 Cap possesses the main neutralizing epitopes of PCV2 and, as a consequence, can induce a strong and protective immune response in pigs [25,26]. The C-terminal region of the PCV2 Cap protein contains the peptide ^224^FNLKDPPLNP^233^ that is part of a conformational epitope that induces virus-neutralizing antibodies [27]. Recently, this C-terminal peptide was inserted into recombinant Papaya ringspot virus (PRSV) coat protein (CP) VLPs. These chimeric VLPs induced high IgG antibody titers in BALB/c mice immunized with a single dose of 10 µg of VLPs adjuvated with Emulsigen-P, an oil-in-water emulsion with polymer [28].

The aim of the present study was to assess the immune response and protective efficacy in 21-day-old pigs following a two-dose schedule vaccination with 25 µg of chimeric VLPs carrying the PCV2 Cap C-terminal peptide, adjuvated with Emulsigen-P, and challenged with a PCV2b Mexican isolate 14 days after the last immunization. PCV2 peptide-specific IgG seroconversion, serum cytokines, viral load in nasal swabs and lymphoid organs, as well as histopathological score, were determined. In line with the findings, the potential use of these VLPs to control PCV2 is discussed.

## 2. Materials and Methods

### 2.1. Vaccine

The vaccine formulation was prepared following the methodology outlined by Her-nández-Aviña et al. [28]. Briefly, a 160 mL culture of *E. coli* BL21 (DE3) strain harboring plasmid pPRSVCP-PCV2-PEP4 was grown at 30 °C, 800 rpm, and at 1 vessel volume/min airflow in a Biotechnology MiniBio V1 bioreactor (Applikon, Delft, The Netherlands). When the culture reached an OD600 = 1, recombinant protein expression was induced with 1 mM IPTG for 20 h. Induced cells were lysed by sonication with an XL-2000 sonicator (Misonix, Farmingdale, NY, USA) in cold lysis buffer (300 mM NaCl, 10 mM Tris, 10 mM EDTA, pH 7.2) supplemented with complete EDTA-free protease inhibitor cocktail (Roche, Basel, Switzerland). Supernatant was taken from the centrifuged lysate and incubated for 1 h at room temperature to allow VLP assembly. Particles were precipitated with 4% PEG 8000 at 350 rpm/1.5 h/4 °C with constant shaking and left for 1 h at room temperature (22 ± 2 °C). Finally, the sample was centrifuged, and the precipitate was solubilized overnight in ¼ *v*/*v* of the initial culture medium of 10 mM sodium phosphate buffer, 300 mM NaCl, 10 mM EDTA. Chimeric VLP concentration was determined with Bradford reagent (Sigma, St. Louis, MI, USA), and purity was assessed by densitometric analysis of SDS-PAGE/Coomassie Blue staining gel in a GelDoc GO Gel Imaging System with Image Lab Software 6.1 (Biorad, Hercules, CA, USA). VLP purity in this batch was 62%. VLPs were not tested for endotoxin levels. Each individual dose consisted of 25 μg of chimeric VLPs harboring the ^224^FNLKDPPLNP^233^ Cap PCV2 peptide, 20% Emulsigen-P adjuvant (MVP adjuvants), and phosphate-buffered saline (PBS) pH 7.4 in a 2 mL volume.

### 2.2. PCV2b Challenge Virus

The challenge virus used in this study was obtained from the inguinal lymph nodes of a weaned pig from a commercial Mexican farm that tested positive for PCV2 using qPCR (GenBank accession number: KT275160).

The lymph nodes were macerated and centrifuged, and the resulting supernatant was filtered through 0.45 and 0.22 µm filters (Merck, Darmstadt, Germany). The supernatant was subjected to DNA extraction with the Tissue DNA Preparation—Column Kit (Jena Bioscience, Jena, Germany), following the manufacturer’s specifications. Viral copy number was determined by quantitative PCR (qPCR) with PCR POCKIT Porcine circovirus type 2 Detection Kit (GeneReach Biotechnology Corporation, Taichung City, Taiwan) in a 30 µL reaction volume and 3 µL of DNA sample, employing a Rotor-Gene Q 5Plex HRM (Qiagen, Venlo, The Netherlands), performing the following program: 40 cycles of 93 °C/15 s and 60 °C/60 s.

### 2.3. Animal Immunization, Challenge, and Sampling

The protocol was approved by the Committee of Facultad de Medicina Veterinaria y Zootecnia (protocol: CICUA 758, approved on 20 October 2024) and was carried out at a commercial farm (Granja FAGA Ixtlahuaca farm, State of Mexico, Mexico).

#### 2.3.1. Animal Immunization

Thirty-three-week-old pigs (*Sus scrofa domesticus*) weighing more than 5 kg were acclimated for five days and randomly allocated to three experimental groups in separate pens of 10 individuals each:

Group A: Immunized with chimeric VLPs carrying Cap PCV2 peptide and challenged. Each pig received two intramuscular injections in the shoulder with 2 mL of the chimeric VLP vaccine formulation. The primary immunization was administered at 26 days of age (day 0), followed by a booster dose at 5 weeks of age (day 14).

Group B: Non-immunized/challenged.

Group C: Non-immunized/unchallenged. This group was housed in a separate pen located 4.5 km away from the other groups to avoid collateral infection.

Sample size was calculated with WinEpi Software 2.0, considering a 65% PCV2 prevalence in the farm. Clinical signs indicating animal distress that were monitored throughout the experiment were coughing, sneezing, skin redness, and dyspnea. Pens were provided with infrared light to keep the temperature between 16 and 24 °C.

#### 2.3.2. Viral Challenge

The group immunized with the chimeric VLPs and a non-immunized group were challenged with PCV2 at 54 days of age (two weeks after the booster dose). Each animal was inoculated intranasally with 1 mL (0.5 mL/nostril) containing 1 × 10^5^ TCID_50_ of PCV2b Mexican isolate, based on a previous study [29].

#### 2.3.3. Sample Collection and Body-Weight Tracking

Blood samples were collected on days 0, 7, 14, 21, 28, 35, 42, and 49. Body weight was measured on the same days as blood sample collection. Nasal swabs were taken on days 0, 14, 28, 35, 42, and 49. The schedule for the sample collection, vaccination, and challenge is shown in Figure 1. Peripheral blood (5–7.5 mL) was collected from the jugular vein in K2 EDTA Vacutainer sterile tubes (Becton Dickinson, Franklin Lakes, NJ, USA), centrifuged at 10,000 rpm/10 min, and serum was transferred to new tubes and stored at −70 °C until use. Sterile Dacron swabs were placed in phosphate-buffered saline (PBS) stored at 4 °C for transport, and frozen at −70 °C upon arrival to the laboratory until further processing for qPCR analysis to assess viral shedding. Two randomly selected individuals per group were euthanized on day 35, and lymph nodes, tonsils, spleen, and lungs were collected for further analysis. The rest of the individuals were euthanized on day 49, and the same specimens were collected from two individuals that were picked randomly per group. All samples and specimens were labeled with blind codes to avoid any bias in data collection.

### 2.4. Antibody Assay

To determine the relative levels of specific antibodies against the PCV2 Cap peptide ^223^FNLKDPPLNP^233^ in the blood sera of all animals, an ELISA assay was carried out. Peptide was synthesized and immobilized in 96-well plates at a concentration of 7–11 pmol/well by JPT Peptide Technologies. The peptide-coated wells were blocked with 200 µL of 1% bovine serum albumin (BSA) in PBS/0.05% Tween 20 (PBST) for 1 h at 37 °C, followed by three PBST washes. Pig serum samples (1:50 dilution in blocking buffer) were added to the wells (200 µL/well). The plate was incubated for 1 h at 37 °C, followed by three washes with PBST. Afterwards, 200 µL of anti-pig IgG–HRP conjugate (Bio-Rad, Hercules, CA, USA) diluted 1:80,000 with PBS/1% BSA was added per well and incubated at 37 °C for 1 h. Finally, 50 μL of 3,3′,5,5′-Tetramethylbenzidine (TMB) substrate was added to each well, the reaction was stopped with 30 μL of 0.2 M sulfuric acid, and absorbance was read at 450 nm without a blank using a Synergy H1 reader (BioTek, Beijing, China).

### 2.5. Cytokine Profiling

To evaluate the modulation of cellular immunity, cytokine levels in serum samples were quantified using the MAGPIX^®^ Multiplexing System (Thermo Fisher Scientific, Waltham, MA, USA). The panel included the anti-inflammatory cytokines interleukin-4 (IL-4), interleukin-10 (IL-10), and interleukin-1 receptor antagonist (IL-1Ra) and pro-inflammatory cytokines interleukin-1α (IL-1α), tumor necrosis factor-α (TNF-α), interferon-γ (IFN-γ), granulocyte-macrophage colony-stimulating factor (GM-CSF), interleukin-8 (IL-8), interleukin-12 (IL-12), interleukin-18 (IL-18), interleukin-1β (IL-1β), interleukin-6 (IL-6), and interleukin-2 (IL-2). Individual samples from each group were pooled before the assay.

### 2.6. PCV2 Genome Presence in Nasal Swabs and Organs by Real-Time PCR

Genomic DNA from nasal swabs and organs was isolated with the Tissue DNA Preparation—Column Kit (Jena Bioscience, Jena, Germany). For DNA isolation from organs, 1 cm^3^ pieces were first processed in the FastPrep-24 5G bead beating grinder and lysis system (MP Biomedicals, Solon, OH, USA) using lysing matrix A tubes and 1 mL of cell lysis solution for tissues and cells. Detection of PCV2 was performed by qPCR using the POCKIT Porcine circovirus type 2 PCR detection kit (GeneReach Biotechnology Corporation, Taichung City, Taiwan), as described in Section 2.2. Samples with Ct values greater than 40 were considered negative.

### 2.7. Histopathological Score

Macroscopical lesions were assessed in all animals post-mortem. The pigs were euthanized in accordance with the Mexican guideline NOM-033-SAG/ZOO-2014. Tonsils, lungs, spleen, and inguinal lymph nodes were taken from two individuals to perform a histopathological study and determine the degree of histological damage. The tissues were preserved in 10% buffered formalin, pH 7.2, and subsequently processed by routine paraffin embedding and hematoxylin/eosin staining techniques [30,31]. The histological sections were evaluated using optical microscopy and assigned to the different injury grades. The lesions that were evaluated in each tissue were determined after performing a rapid assessment, in which the most relevant changes were identified. Lesions were classified based on severity and distribution as mild (minimal tissue damage with limited vascular changes and exudate), moderate (evident tissue damage with leukocyte infiltration and vascular alterations), or severe (extensive tissue destruction with abundant exudate).

### 2.8. Statistical Analysis

Antibody titers were analyzed using chi-square (χ^2^) tests. Analysis of variance (ANOVA) was applied to evaluate differences in initial body weight and average daily gain among groups using the SAS-JMP Pro 16 statistical software. Where significant differences were detected, Tukey’s HSD test was employed for post hoc comparisons. No animals or data points were excluded in the analysis.

## 3. Results

### 3.1. VLPs Carrying a PCV2 Peptide Induce Peptide-Specific IgG Antibodies in Immunized Animals Under a Prime-BOOST Scheme

Serum IgG response against the PCV2 Cap C-terminal peptide in pigs immunized with the chimeric VLPs on days 0 and 14 was followed. A modest increase in IgG titers seven days after the first immunization (day 7) was observed, while the highest titers occurred 14 days after the second immunization (day 28) and 7 days after viral challenge (day 35), gradually dropping on days 14 and 21 after challenge (days 42 and 49). In contrast, IgG titers in non-immunized/unchallenged pigs remained negligible at all time points. Lastly, IgG levels in non-immunized/challenged pigs were low before challenge, rose on day 7 after challenge, and underwent a sharp decrease on days 14 and 21 after challenge (Figure 2).

### 3.2. VLPs Carrying a PCV2 Peptide Induce Anti-Inflammatory/Pro-Inflammatory Cytokine Production in Immunized Animals

To investigate whether the adjuvated VLPs carrying the PCV2 peptide modulate the immune response in vaccinated animals, the serum concentration of several cytokines was measured. In regard to anti-inflammatory cytokines, IL-4, IL-10, and IL-1Ra increased 7 days after viral challenge. As for pro-inflammatory cytokines, IL-1α, TNF-α, IL-8, IL-1β, IL-2, and GM-CSF levels were low at all time points, IL-12 and IL-18 levels were elevated 7 days after booster and increased 7 days after viral challenge, IFN-γ was high 7 days after booster only and, finally, IL-6 had a very gentle increase 7 days after challenge. It is important to note that there were considerable amounts of some cytokines in the non-immunized groups. For example, IL-12 and IL-18 levels were quite similar to the immunized group at both times; IL-4 was much higher in the unchallenged group and IFN-γ levels in the non-immunized group before challenge (Table 1).

### 3.3. VLPs Carrying a PCV2 Peptide Reduce Viral Shedding and Presence in Tonsils and Lungs of Immunized Animals Challenged with a PCV2 Isolate

Assessment of PCV2 viremia in challenged groups by qPCR showed that nasal swabs from all the animals belonging to the non-immunized group were PCV2-positive on days 7, 14, and 21 after challenge. On the contrary, PCV2 was detected in one sample of the immunized group on day 7 after challenge, and two individuals tested positive on days 14 and 21. All nasal swabs from the non-immunized/unchallenged group were PCV2-negative at all time points. It is worth mentioning that all samples collected from the three experimental groups at days 0, 14, and 28 post-immunization were negative for PCV2. PCV2 genomes were detected in the lungs of six animals from the non-immunized group and four animals from the immunized group 21 days after challenge. Finally, tonsils of six animals from the non-immunized group and two animals from the immunized group were PCV2-positive 21 days post-challenge (Figure 3). Statistically significant differences in Ct values obtained from nasal swabs were detected between the immunized and non-immunized groups on days 7, 14, and 21 post-challenge (Appendix A).

### 3.4. VLPs Carrying a PCV2 Peptide Showed No Effect on Daily Weight Gain and Macroscopic and Histopathological Lesions in Lungs, Spleens, Tonsils, and Lymph Nodes of Immunized Animals Challenged with a PCV2 Isolate

The body weights of all the individuals were followed throughout the experiment (Table 2). The non-immunized/unchallenged group showed the highest daily weight gain, and there was no difference between the challenged groups.

The protection against PCV2-associated lesions in pigs immunized with chimeric VLPs carrying the PCV2 Cap C-terminal peptide was evaluated (Table 3). Tonsil and inguinal lymphadenomegalia and edematous interlobular septa (macroscopic lesions) were found in two and all individuals, respectively. No signs of tonsil and inguinal lymphadenomegalia were present in the non-immunized groups, while nine and eight individuals from the non-immunized/challenged and non-immunized/unchallenged groups, respectively, had edematous interlobular septa.

Regarding histopathological lesions, VLP-immunized/challenged pigs displayed the following: mild-to-moderate lymphocytic interstitial pneumonia and class 2 pneumonia; mild follicular hyperplasia, mild-to-moderate lymphoid hyperplasia in spleen, mild-to-moderate follicular hyperplasia, and moderate-to-severe lymphoid hyperplasia; and moderate necrosis in tonsils and inguinal lymph nodes. Non-immunized/challenged pigs presented with the following: mild lymphocytic interstitial pneumonia and class 2 pneumonia; moderate-to-severe follicular hyperplasia, mild lymphoid hyperplasia in spleen, moderate follicular hyperplasia, and moderate lymphoid hyperplasia; and moderate necrosis in tonsils and inguinal lymph nodes. Finally, non-immunized/unchallenged individuals exhibited the following: mild lymphocytic interstitial pneumonia; mild lymphoid hyperplasia in spleen, mild follicular hyperplasia, and mild lymphoid hyperplasia in tonsils and inguinal lymph nodes (the most representative histopathological lesions are shown in Appendix A).

## 4. Discussion

The search for new protective vaccines that are both economical, accessible, and cost-effective is attributed to advances in process analytical technology, quality by design, experimental design, and bioprocessing that allow for convenient and economical production. Furthermore, recombinant vaccines are expected to stimulate a robust humoral immune response, reduce or eliminate clinical symptoms and viral shedding, and prevent target organ damage [32]. The present work aimed to evaluate the protective efficacy of an experimental vaccine based on chimeric VLPs carrying a PCV2 Cap C-terminal peptide in its target species. To evaluate this experimental vaccine, an intranasal PCV2 challenge dose of 1 × 10^5^ TCID_50_ was employed two weeks after booster according to previous research [29]. Although the intranasal route has been employed for PCV2 challenge [15], intranasal/intramuscular inoculation is more common to achieve PCV2 clinical infection [3,14,24,26].

Most of the studies on PCV2 focus on assessing the generation of antibodies through ELISA due to its high sensitivity and specificity [33,34,35]. In this study, the serum samples from animals immunized with the chimeric VLP-based vaccine had higher antibody titers against the PCV2 Cap C-terminal peptide compared to serum samples from non-immunized animals, although statistically significant differences were found 7 and 28 days after priming, and 7, 14, and 21 days post-challenge (35, 42, and 49 days post-immunization). The changes in antibody titers in the chimeric VLP-immunized group indicated that the first dose triggered an initial response, and the second one stimulated a robust humoral immune response, which was maintained up to 21 days after vaccine priming. Finally, the antibody levels declined 14 and 21 days post-challenge, perhaps due to their recruitment into lymphoid organs for viral control, suggesting that the antibodies elicited by the chimeric VLP vaccine are sensitive and specific to the PCV2b isolate used for the challenge. This finding is highly relevant, as a previous study showed that the PCV2b is the most prevailing genotype in Mexico [36].

It has been found that IFN-γ plays an important role in PCV2 control. A robust study that assessed the efficacy of four commercial PCV2 vaccines reports an inverse correlation between the number of IFN-γ-secreting peripheral blood mononuclear cells and blood PCV2 viral load in vaccinated/challenged and unvaccinated/challenged groups [37]. In this work, a type 1 (Th1) immune response in the immunized group was predicted, as the adjuvant employed in the chimeric VLP formulation (Emulsigen-P) stimulates IFN-γ and IL-12 secretion. Unexpectedly, IFN-γ concentration in the blood of the vaccinated group was unexplainably lower than that found in one of the unvaccinated groups. Albeit IL-12 and IL-18, which in turn stimulate IFN-γ production, were induced 21 days after vaccine priming, considerable levels of these cytokines were also present in the non-immunized groups. An alternative assay, such as quantifying cytokines—in particular IFN-γ—in re-stimulated peripheral blood mononuclear cells isolated from the blood samples that were collected in all groups, might have been more appropriate for evaluating the specific immune response induced by the chimeric VLPs in order to make a clear correlation between cytokine profile and viral load, for instance.

A relevant finding of this study is the reduction in PCV2 viral shedding in nasal swabs from the group immunized with the chimeric VLP-based vaccine. The oronasal route has been reported as the main route of transmission of PCV2 [38], and is a determining factor of the disease because the virus can be shed up to 70 days after infection [39]. All the nasal swabs taken from the unvaccinated/challenged group were PCV2-positive 7, 14, and 21 days after challenge, while 10% of the samples from the vaccinated group were positive 7 days after challenge and reached 25% on days 14 and 21. In addition, there was a reduction in viral presence in lungs and tonsils, indicating that the chimeric VLP-based vaccine mitigates PCV2 infection.

PCV2 infection causes growth retardation [40]; however, in the present study, there were no statistically significant differences in weight gain between vaccinated and unvaccinated groups challenged with the PCV2 isolate. Regarding macroscopic and histopathological lesions, these were practically the same in both groups, but lesions were also found in the unchallenged group. Our hypothesis is that, since the farm of origin is not a controlled one, the piglets were likely exposed to various endemic pathogens prior to entering this study. It is well established that PCV2-associated disease often occurs in the context of coinfections, and several pathogens commonly circulating in commercial herds—such as *Actinobacillus pleuropneumoniae*, *Mycoplasma hyopneumoniae*, *Streptococcus suis*, and *Pasteurella multocida*—can cause respiratory or lymphoid lesions that overlap with those typically attributed to PCV2 infection. Such background infections could explain the presence of histopathological changes in the non-immunized/unchallenged group, even in the absence of experimental PCV2 challenge.

Taken together, the chimeric VLP vaccine carrying a PCV2 Cap C-terminal peptide reduces viral shedding without an evident reduction in histopathological lesions under commercial farm conditions. To clarify this, an additional study with pathogen-free individuals in a controlled environment needs to be carried out. Additionally, a higher VLP dose and alternative immunization routes to enhance mucosal immunity could be considered. Lastly, long-term reduction in viral shedding must be evaluated in a further study, and it is possible that more than two immunizations would be needed for a long-term viral reduction, although this could be impractical in the specific case of PCV2, as commercial vaccines for controlling this virus are administered once [37].

## 5. Conclusions

This study shows the efficacy of a new experimental vaccine containing a 25 μg/dose of filamentous VLPs carrying a PCV2 Cap C-terminal peptide in pigs. This vaccine induced PCV2-specific antibodies in immunized animals under a prime/boost schedule. Finally, PCV2 replication was significantly reduced in these animals after viral challenge with a PCV2b Mexican field isolate, suggesting that this experimental vaccine may be useful for controlling viral shedding in pig production regions where this is the most prevalent genotype.

## Figures and Tables

**Figure 1 vaccines-14-00024-f001:**
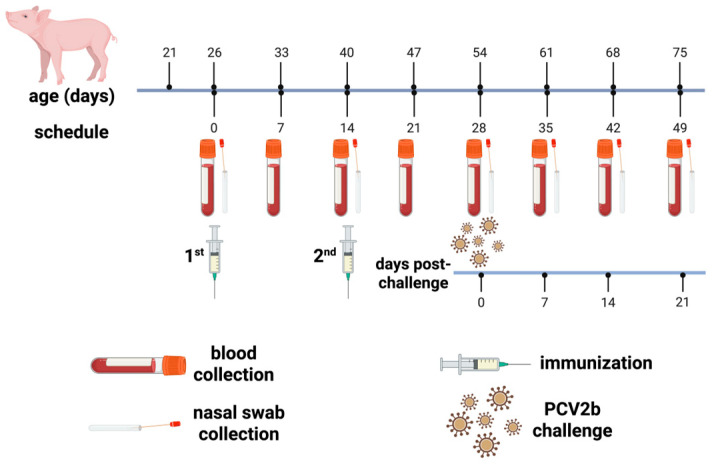
Sample collection, immunization, and viral challenge schedule for the animal protocol. The chimeric VLP-based vaccine was administered intramuscularly. For viral challenge, 1 × 10^5^ TCID_50_ of PCV2b Mexican isolate was inoculated intranasally. Created in Biorender (Science Suite Inc., Toronto, ON, Canada).

**Figure 2 vaccines-14-00024-f002:**
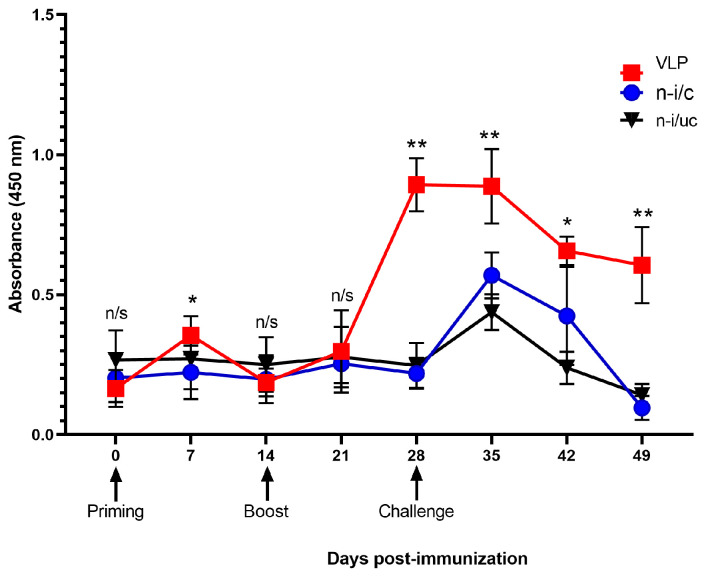
IgG antibody response in experimental groups after immunization and challenge. VLP: group immunized with chimeric VLPs carrying PCV2 Cap carboxy-terminal peptide; n-i/c: non-immunized/challenged group; n-i/uc: non-immunized/unchallenged group. Priming, boost immunization, and challenge days are shown with arrows. Statistically significant differences between the immunized group and the other groups were detected on days 7, 28, 35, 42, and 49, as determined by multiple *t*-test analysis. Significance is indicated as * *p* < 0.001 and ** *p* < 0.0001, n/s, not significant. Sample size was *n* = 10 for all time points, except for days 42 and 49 (*n* = 8).

**Figure 3 vaccines-14-00024-f003:**
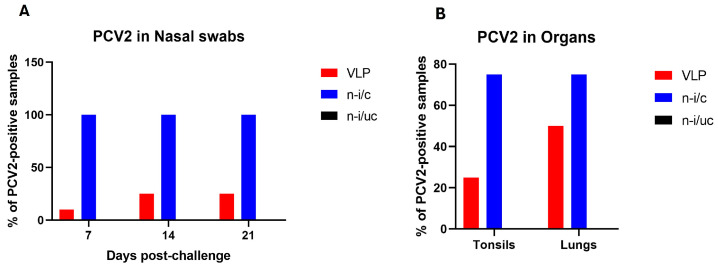
Percentage of PCV2-positive samples detected in experimental groups: (**A**) Percentage of PCV2-positive nasal swab samples at 7, 14, and 21 days post-challenge (d.p.c.) in the non-immunized/challenged (n-i/c), VLP-immunized, and non-immunized/unchallenged (n-i/uc) groups. (**B**) Percentage of PCV2-positive tonsil and lung samples collected at 21 d.p.c. in the same groups. Detection of PCV2 was performed by quantitative PCR (qPCR), and data are presented as the percentage of positive samples within each group (*n* = 8). Samples with a Ct value greater than 40 were considered negative.

**Table 1 vaccines-14-00024-t001:** Cytokine levels in serum samples taken 21 and 35 days after vaccine priming.

Cytokine	Groups					
	Immunized/Challenged (µg/mL)		Non-Immunized/Challenged (µg/mL)		Non-Immunized/Unchallenged (µg/mL)	
	21d	35d	21d	35d	21d	35d
IL-4	<61	341.8	<61	62	<61	1011.35
IL-10	<24	252.1	<24	<24	<24	48.6
IL-1Ra	140.5	262.6	42.1	176.2	24.2	110.5
IL-1α	4.2	17.6	9.8	14.2	3.8	27.3
TNF-α	<24	<24	79.4	<24	<24	<24
IFN-γ	855.7	<122	1499	<122	<122	<122
GM-CSF	<24	<24	<24	<24	<24	<24
IL-8	<12	<12	<12	<12	<12	164.9
IL-12	437.3	555.8	216.7	262.8	145.1	550
IL-18	299.6	518.9	153.5	178.6	165.7	347.4
IL-1β	<122	<122	<122	<122	<122	<122
IL-6	<24	43.3	<24	<24	<24	<24
IL-2	<24	<24	<24	<24	<24	39.1

**Table 2 vaccines-14-00024-t002:** Weight gain in the experimental groups.

	Groups		
	Immunized/Challenged	Non-Immunized/Challenged	Non-Immunized/Unchallenged
initial weight (day 0) (kg)	8.98 ± 0.98	9.76 ± 1.27	8.9 ± 1.21
final weight (day 49) (kg)	38.63 ± 3.49	39.08 ± 3.36	43.65 ± 3.49
weight gain (kg)	29.65	29.32	34.75
daily weight gain (kg)	0.54 ± 0.07 ^a^	0.54 ± 0.07 ^a^	0.62 ± 0.10 ^b^

Different letters (^a^,^b^) indicate statistically significant differences (*p* < 0.05). *n* = 10 and 8 on initial and final weight, respectively, for all groups.

**Table 3 vaccines-14-00024-t003:** Macroscopic and histopathological lesions in lungs, spleens, tonsils, and lymph nodes in two animals from each experimental group.

Lesion	Groups		
	Immunized/Challenged	Non-Immunized/Challenged	Non-Immunized/Unchallenged
tonsil and inguinal lymphadenomegalia *	2/10	0/10	0/10
edematous interlobular septa *	10/10	9/10	8/10
lymphocytic interstitial pneumonia	+, ++	+, +	+, +
class 2 pneumonia	+, ++	+, +	-, -
spleen follicular hyperplasia	+, +	++, +++	-, -
spleen lymphoid hyperplasia	+, ++	+, +	+, +
tonsil and inguinal lymph node follicular hyperplasia	+, ++	++, ++	+, +
tonsil and inguinal lymph node lymphoid hyperplasia	++, +++	++, ++	+, +
tonsil and inguinal lymph node necrosis	++, ++	++, ++	-, -

-: no lesion; +: mild; ++: moderate; +++: severe. *n* = 2 for histopathological lesions. * the number of animals presenting macroscopic lesions/total animals examined is reported.

## Data Availability

The original contributions presented in this study are included in the article. Further inquiries can be directed to the corresponding authors.

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
