# Peer review of "Virus-like Particles Carrying a Porcine Circovirus Type 2b Peptide Induce an Antibody Response and Reduce Viral Load in Immunized Pigs"

_vaccines, 2025, doi:10.3390/vaccines14010024_

Round 1
Reviewer 1 Report
Comments and Suggestions for Authors
Comments:
The manuscript by Hernández-Aviña et al. described an experimental study evaluating the immunogenicity and protective efficacy of VLPs expressing a C-terminal peptide from the PCV2b capsid protein, produced in E. coli and adjuvanted with Emulsigen-P. The VLPs were tested in a prime-boost immunization scheme in 3-week-old pigs, followed by intranasal challenge with a Mexican PCV2b isolate. This study reported modest IgG induction peaking at day 28 post-immunization, some elevation in pro- and anti-inflammatory cytokines, reduced viral shedding in nasal swabs and select tissues compared to non-immunized controls, but no significant improvements in weight gain or lesion severity. While the topic is relevant given ongoing genotype shifts in PCV2 and the need for updated vaccines, I still have critical concerns to be addressed.
Major concerns:
- The core claim of "protective efficacy" is not supported by the results. While reduced viral shedding in nasal swabs (1/10 vs. 10/10 positive at 7 days post-challenge [dpc]) and tissues (e.g., 2/8 vs. 6/8 positive tonsils at 21 dpc) is noted, this is not accompanied by meaningful clinical or pathological protection. Histopathological lesions were equally severe or even worse in immunized pigs for several parameters (e.g., moderate-severe lymphoid hyperplasia and necrosis in tonsils/lymph nodes; Table 4). Macroscopic lesions (e.g., lymphadenomegaly) were also even more prevalent in the immunized group (2/10 vs. 0/10). The abstract and discussion acknowledge that "histological lesions were not less severe" yet pivot to a tentative endorsement of the vaccine for shedding control. This disconnect undermines the manuscript's validity—shedding reduction alone, without lesion mitigation or growth performance benefits, does not constitute "efficacy" in a production disease context like PCVAD.
- The vaccine includes 20% Emulsigen-P, a potent adjuvant known to induce non-specific immunity. Without an adjuvant-only group, it's impossible to attribute responses (e.g., cytokine shifts or IgG) to the VLPs versus the adjuvant. This is a critical omission for a subunit vaccine study.
- The PCV2b isolate is sourced from a field sample and titered via a commercial kit, but no full genomic sequencing, purity assessment (e.g., co-infecting agents), or in vitro replication kinetics are provided. The intranasal route (1 mL split?) is unconventional for PCV2 (typically intramuscular/oral for PCVAD models) and may not mimic natural exposure.
- Many key results lack statistical testing. For example, (1) viral shedding and tissue loads (Table 2) report raw positives but no chi-square/Fisher's exact tests for group differences; (2) cytokine data (Table 1) are pooled and unanalyzed—e.g., no ANOVA or t-tests for IL-12/IL-18 elevations.
Minor concerns:
- Suggest adding RT-qPCR results of virus shedding and Figures of histopathological lesions.
- Suggest adding error bars and exact p-values. Legend arrows for immunization/challenge are unclear.
- Language should be fully proofread. E.g., "weaned pig located in a Mexican farm" should be "from a Mexican farm." Use consistent tense (past for methods/results).
Author Response
Comments:
The manuscript by Hernández-Aviña et al. described an experimental study evaluating the immunogenicity and protective efficacy of VLPs expressing a C-terminal peptide from the PCV2b capsid protein, produced in E. coli and adjuvanted with Emulsigen-P. The VLPs were tested in a prime-boost immunization scheme in 3-week-old pigs, followed by intranasal challenge with a Mexican PCV2b isolate. This study reported modest IgG induction peaking at day 28 post-immunization, some elevation in pro- and anti-inflammatory cytokines, reduced viral shedding in nasal swabs and select tissues compared to non-immunized controls, but no significant improvements in weight gain or lesion severity. While the topic is relevant given ongoing genotype shifts in PCV2 and the need for updated vaccines, I still have critical concerns to be addressed.
Major concerns:
1.- The core claim of "protective efficacy" is not supported by the results. While reduced viral shedding in nasal swabs (1/10 vs. 10/10 positive at 7 days post-challenge [dpc]) and tissues (e.g., 2/8 vs. 6/8 positive tonsils at 21 dpc) is noted, this is not accompanied by meaningful clinical or pathological protection. Histopathological lesions were equally severe or even worse in immunized pigs for several parameters (e.g., moderate-severe lymphoid hyperplasia and necrosis in tonsils/lymph nodes; Table 4). Macroscopic lesions (e.g., lymphadenomegaly) were also even more prevalent in the immunized group (2/10 vs. 0/10). The abstract and discussion acknowledge that "histological lesions were not less severe" yet pivot to a tentative endorsement of the vaccine for shedding control. This disconnect undermines the manuscript's validity—shedding reduction alone, without lesion mitigation or growth performance benefits, does not constitute "efficacy" in a production disease context like PCVAD.
RESPONSE: We totally agree with this concern and have changed the title “Immunogenicity and protective efficacy of E. coli-expressed virus-like particles carrying a porcine circovirus type 2b peptide” to “Virus-like particles carrying a porcine circovirus type 2b peptide induce an antibody response and reduce viral load in immunized pigs”, highlighting the most relevant results shown in the manuscript.
2.- The vaccine includes 20% Emulsigen-P, a potent adjuvant known to induce non-specific immunity. Without an adjuvant-only group, it's impossible to attribute responses (e.g., cytokine shifts or IgG) to the VLPs versus the adjuvant. This is a critical omission for a subunit vaccine study.
RESPONSE: We truly appreciate the comment. Regarding IgG response, it is unlikely but not impossible that the adjuvant by itself elicits a specific response towards a PCV2 peptide. With respect to cytokines, we think that an in vitro assay using peripheral blood mononuclear cells re-stimulated with the VLPs and measuring cytokine secretion would have been more appropriate to address the role of the VLP, not the adjuvant, in the cytokine response.
3.- The PCV2b isolate is sourced from a field sample and titered via a commercial kit, but no full genomic sequencing, purity assessment (e.g., co-infecting agents), or in vitro replication kinetics are provided. The intranasal route (1 mL split?) is unconventional for PCV2 (typically intramuscular/oral for PCVAD models) and may not mimic natural exposure.
RESPONSE: We thank the reviewer’s comments. We have observed that PCV2 is not easily isolated in vitro and can lose pathogenicity when adapted to cell lines. As reviewer mentions, the most common way to inoculate PCV2 for clinical studies is the intranasal/intramuscular route, but we adopted the method reported by Blanchard et al. Details of the immunization route have been clarified in manuscript. The full reference is: Blanchard P.; Mahé, D.; Cariolet, R.; Keranflec'h, A.; Baudouard, M.A.; Cordioli, P.; Albina, E.; Jestin, A. Protection of swine against post-weaning multisystemic wasting syndrome (PMWS) by porcine circovirus type 2 (PCV2) proteins. Vaccine 2003, 21, 4565-4575. doi: 10.1016/s0264-410x(03)00503-6
The PCV2 isolate used in this study was partially sequenced and corresponds to the KT275160 EdoMex.Mex-13 isolate (Genebank Access number: KT275160). Virus sequencing was performed as follows: Virus used for challenge was sequenced using an ORF2 reading. The PCR reaction was conducted using the DNA extracted and a commercial kit (Biotecmol) following the manufacturer’s instructions. The primers used for the first PCR reaction were: forward (3´ CTGATTACCAGCAATCAGACC 5´) and reverse (3´ ACCGYTACCGYTGGAGAAG 5’), to obtain an 831pb segment. The reaction was carried out under the following conditions in a BIO-RAD thermal cycler: one cycle of 95°C during 5; 35 cycles of 94°C for 35 s, 60°C for 35 s and 72°C for 35 s. Finally, one cycle of 72°C for 10 min was applied. Sequencer ABI PRISM 310 Genetic Analyzer (PE Applied Biosystems, Foster City, USA) was used for sequence Reading. This isolate is reported in Bedolla-López F., et. al., (2018), which is cited as reference 36 in the manuscript.
4.- Many key results lack statistical testing. For example, (1) viral shedding and tissue loads (Table 2) report raw positives but no chi-square/Fisher's exact tests for group differences; (2) cytokine data (Table 1) are pooled and unanalyzed—e.g., no ANOVA or t-tests for IL-12/IL-18 elevations.
RESPONSE: We thank the reviewer for this important comment. We agree that statistical testing would normally strengthen conclusions derived from viral load and cytokine measurements. However, in this study the samples used for cytokine quantification (and in some cases for viral detection) were processed as pooled sera per group, rather than as individual-animal replicates. This experimental design allowed us to obtain sufficient volume for the multiplex cytokine assay but, inherently, does not allow for statistical comparisons, as the pool represents a single composite measurement per group and time-point. For this reason, we report the raw concentration values in Table 1, which are intended to illustrate the general immunological trends observed after immunization and challenge, while avoiding overinterpretation. We have now clarified this limitation in the Methods and Results sections to prevent potential misunderstanding.
Minor concerns:
1.- Suggest adding RT-qPCR results of virus shedding and Figures of histopathological lesions.
RESPONSE: We appreciate the reviewer’s suggestion. Real time PCR data of virus shedding have now been incorporated in Supplementary Figure 1, depicting Ct values in nasal swabs and organs. As detailed in the figure legend, negative qPCR samples were uniformly assigned a Ct value of 40 to allow statistical analysis. We decided not to include histopathological images, because we did not found a clear difference between immunized and non-immunized groups.
2.- Suggest adding error bars and exact p-values. Legend arrows for immunization/challenge are unclear.
RESPONSE: We thank the reviewer for these helpful recommendations. Error bars have been added to all relevant figures, and statistical significance is now indicated directly on the graphs. The statistical analysis used (one-way ANOVA and multiple variable t test analysis were performed) is described in corresponding figure legend. Exact p-values have been included where applicable. Additionally, the arrows in the figure illustrating the immunization and challenge schedule have been redrawn and relabeled to improve clarity.
3.- Language should be fully proofread. E.g., "weaned pig located in a Mexican farm" should be "from a Mexican farm." Use consistent tense (past for methods/results).
RESPONSE: We thank the reviewer for this observation. The entire manuscript has been carefully proofread to correct grammatical issues and ensure consistent use of past tense throughout the Methods and Results sections. The phrase “weaned pig located in a Mexican farm” has been changed to “weaned pig from a Mexican farm,” and all similar constructions have been corrected. All language-related corrections have now been incorporated in the revised manuscript.
Reviewer 2 Report
Comments and Suggestions for Authors
Dear editor,
The manuscript entitled "Immunogenicity and protective efficacy of E. coli-expressed virus-like particles carrying a porcine circovirus type 2b peptide" By Dr. Ana del Socorro Hernández-Aviña et al., discusses the out come of using E.coli expressed virus-like particles (VLPs) carrying different PCV2 capsid (Cap) peptides on protecting suseptible animals (pigs) from PCV2 infection using an isolate that is circulating in Mexico. This came after showing that those particles can induce immune reponse in BALB/c mice after a single subcutaneous injection. In my opinion the reseach is fine, but it can be modified a little bit, and will be ready for publication. The author can find my suggestions in the PDF file that I uploaded.

Author Response
Although numerous studies related to the development of PCV2 vaccines have been conducted by various researchers, the relevance of this topic remains. The advantage of this study is the use of a challenge, as most studies assess vaccine efficacy only based on the level of specific antibodies. In my opinion, this manuscript deserves publication. I believe it would be interesting to compare its effectiveness with the commercial vaccines most commonly used in Mexico in the future.
1.- please rephrase this sentence (Line 35)
RESPONSE: Thanks, sentence has been rewritten.
2.-You can add a schematic figure here showing the process (by days) of immunzation, challenge and sampling
RESPONSE: We appreciate the reviewer’s suggestion. A new schematic figure has been added to the revised manuscript to illustrate the timeline of immunization, challenge, sample collection, and subsequent analyses. This figure highlights the key experimental steps and improves the overall clarity of the study design.
3.- this article can also be mentioned here:
A Triple Gene-Deleted Pseudorabies Virus-Vectored Subunit PCV2b and CSFV Vaccine Protects Pigs against PCV2b Challenge and Induces Serum Neutralizing Antibody Response against CSFV
February 2022Vaccines 10(2):305
DOI: 10.3390/vaccines10020305
LicenseCC BY 4.0
RESPONSE: We thank the reviewer for this helpful recommendation. The suggested article has been incorporated into the revised manuscript in the corresponding section.
4.- were the animals tested before the start of the experiment for PCV or other animal diseases? did you start the experiment on the same of housing the animals or did you leave them for adaptation first.
RESPONSE: The pigs were taken to the experimental pens at 21 days of age where they were acclimated for 5 days. On day 26, blood and nasal swab sampling was performed in the 3 groups and all pigs were negative using the POCKIT PCV2 Detection Kit for Porcine Circovirus Type 2 Detection (GeneReach Biotechnology Corporation).
5.- can you please use different colours in the graph?
RESPONSE: Figure 1 has been updated to enhance visual differentiation among the groups by using distinct colors. Red squares correspond to the VLP-immunized group; blue circles represent the non-immunized/challenged (n-i/c) group; and black inverted triangles represent the non-immunized/unchallenged (n-i/uc) group. In addition, statistical significance is now indicated above the corresponding time points.
6.- How come the n-i/uc group also had an IgG antibody response on 35th day?
RESPONSE: We appreciate this observation. We did not expect IgG levels to rise in this group throughout all time-points. We do not know what caused this. Nonetheless, this increase on day 35 was significantly lower than the immunized group.
Reviewer 3 Report
Comments and Suggestions for Authors
Hernández-Aviña et al, set out to determine the efficacy of a chimaeric VLP vaccine against PCV2 which has been produced in E. coli. In contrast to previous studies, they test the vaccine candidate in the target species and measure the serological response, cytokine responses across the timeline of the study as well as viral load and histopathological phenotypes following viral challenge. The authors present a well-characterised study, although the results do not necessarily suggest that these VLPs could be used as an effective vaccine. Below are some points that would both strengthen and improve the manuscript for the readership.
Major comments
The VLP vaccine has been produced in E. coli. I was wondering if the authors have tested their final formulations for the presence of endotoxin or LPS and if so, whether this could have had an influence on some of the responses detailed within this manuscript?
Figure 1 - The authors state there is a statistically significant difference between the ELISA results on Day 28, 35 and 49. Could the authors report the p-values either on the graph or in the text, whilst also denoting on the graph the statistically significant points using an * or similar. Additionally, could the authors please describe the statistical analysis performed for this data in the figure legend.
Section 3.2 - Could the authors provide a more detailed section highlighting the key differences between vaccinated and unvaccinated animals, pre and post-challenge? Currently this section provides a list of the cytokine levels present in vaccinated animals but no obvious analysis or conclusions are provided.
Section 3.4 - Were the authors expecting the histopathological report to highlight lesions in the unchallenged/unimmunised groups? Could the authors provide a possible explanation for this?
Conclusions - Whilst the authors show the the VLP vaccine reduces viral shedding after challenge, this is also common for numerous vaccines at early time points post-vaccination and this aspect of protection tends to wane over time. Perhaps the authors could reflect on this in their discussion as to how often they would assume the livestock would need to be immunised to provide protection? Additionally, this study did not show that this vaccine reduced disease in these animals, could the authors provide a hypothesis as to whether they think this means this vaccine is unlikely to be effective in the real world, or perhaps suggest some future experiments/trials that may prove the efficacy of this vaccine e.g differing dosing or immunisation site?
Minor Comments
The authors do not describe the production and purification of the VLPs in the methods section, whilst it refers back to a previous manuscript, it would be beneficial to the readership to get a brief description of this process.
Line 283 - Could the authors provide a hypothesis or attempt an explanation for the reduced IFn gamma levels in the vaccinated cohort?
Line 285 - Should read '21 days' after vaccine priming
Author Response
Hernández-Aviña et al, set out to determine the efficacy of a chimaeric VLP vaccine against PCV2 which has been produced in E. coli. In contrast to previous studies, they test the vaccine candidate in the target species and measure the serological response, cytokine responses across the timeline of the study as well as viral load and histopathological phenotypes following viral challenge. The authors present a well-characterised study, although the results do not necessarily suggest that these VLPs could be used as an effective vaccine. Below are some points that would both strengthen and improve the manuscript for the readership.
Major comments
The VLP vaccine has been produced in E. coli. I was wondering if the authors have tested their final formulations for the presence of endotoxin or LPS and if so, whether this could have had an influence on some of the responses detailed within this manuscript?
RESPONSE: We truly appreciate the comment. Actually, the VLPs are enriched through PEG8000 precipitation. This reagent does not precipitate LPS. A description of expression and purification of VLPs has been included in section 2.1. Nonetheless, it would have been important to assess LPS levels in the VLP batch.
Figure 1 - The authors state there is a statistically significant difference between the ELISA results on Day 28, 35 and 49. Could the authors report the p-values either on the graph or in the text, whilst also denoting on the graph the statistically significant points using an * or similar. Additionally, could the authors please describe the statistical analysis performed for this data in the figure legend.
RESPONSE: Figure 1 has been updated to clearly indicate the statistical significance between groups. The corresponding p-values are now displayed using asterisks above the relevant time points (*p < 0.001; **p < 0.0001). In addition, the figure legend has been expanded to describe the statistical analysis performed, including the test used and the criteria for significance.
Section 3.2 - Could the authors provide a more detailed section highlighting the key differences between vaccinated and unvaccinated animals, pre and post-challenge? Currently this section provides a list of the cytokine levels present in vaccinated animals but no obvious analysis or conclusions are provided.
RESPONSE: Thanks for the suggestion. A new Figure (Figure 1) has been included in order to clarify this.
Section 3.4 - Were the authors expecting the histopathological report to highlight lesions in the unchallenged/unimmunised groups? Could the authors provide a possible explanation for this?
RESPONSE: We appreciate the reviewer’s insightful question. Our working hypothesis is that the since the farm of origin was not a controlled one, the piglets were likely exposed to various endemic pathogens prior to entering the study. It is well established that PCV2-associated disease often occurs in the context of coinfections, and several pathogens commonly circulating in commercial herds including Actinobacillus pleuropneumoniae, Mycoplasma hyopneumoniae, Streptococcus suis, and Pasteurella multocida can induce respiratory or lymphoid lesions that overlap with those typically attributed to PCV2 infection. Such background infections could explain the presence of histopathological changes in the unvaccinated, unchallenged group, even in the absence of experimental PCV2 challenge. This interpretation is consistent with the fact that the animals originated from an unvaccinated farm and were not maintained under pathogen-controlled conditions, increasing the likelihood of prior exposure to multiple agents capable of producing similar lesions.
Conclusions - Whilst the authors show the the VLP vaccine reduces viral shedding after challenge, this is also common for numerous vaccines at early time points post-vaccination and this aspect of protection tends to wane over time. Perhaps the authors could reflect on this in their discussion as to how often they would assume the livestock would need to be immunised to provide protection? Additionally, this study did not show that this vaccine reduced disease in these animals, could the authors provide a hypothesis as to whether they think this means this vaccine is unlikely to be effective in the real world, or perhaps suggest some future experiments/trials that may prove the efficacy of this vaccine e.g differing dosing or immunisation site?
RESPONSE: The results of this study demonstrate that vaccination reduces viral shedding, however, it was not shown to reduce the disease. Regarding the expected duration of protection, our study did not evaluate long-term immunity; however, based on the behavior of existing PCV2 vaccines, protection would be expected to wane over time. Future studies assessing antibody persistence and extended challenge periods will be necessary to estimate an appropriate re-vaccination interval. Although this study did not demonstrate a reduction in clinical disease, we do not interpret this as evidence of limited field effectiveness. Instead, the subclinical nature of infection likely masked clinical differences. Additional trials including different dosing strategies, alternative immunization sites, or field studies under natural coinfection pressures would help more clearly determine the vaccine’s efficacy under field conditions.
Minor Comments
The authors do not describe the production and purification of the VLPs in the methods section, whilst it refers back to a previous manuscript, it would be beneficial to the readership to get a brief description of this process.
RESPONSE: We thank the reviewer for this helpful comment. A concise description of the VLP production and purification process has now been added to the Methods section. Although the protocol was previously published, we agree that including a brief summary improves clarity for the readership. All corresponding updates have been incorporated into the revised manuscript.
Line 283 - Could the authors provide a hypothesis or attempt an explanation for the reduced IFn gamma levels in the vaccinated cohort?
RESPONSE: That is such a good observation. We do not have a possible explanation for this. We have to take into account that the cytokine levels measured in the blood samples might be due to several stimuli in the farm. Perhaps, an in vitro assay using peripheral blood mononuclear cells re-stimulated with the VLPs and measuring cytokine secretion would have been more appropriate.
Line 285 - Should read '21 days' after vaccine priming
RESPONSE: Change has been made.
Reviewer 4 Report
Comments and Suggestions for Authors
Although numerous studies related to the development of PCV2 vaccines have been conducted by various researchers, the relevance of this topic remains. The advantage of this study is the use of a challenge, as most studies assess vaccine efficacy only based on the level of specific antibodies. In my opinion, this manuscript deserves publication. I believe it would be interesting to compare its effectiveness with the commercial vaccines most commonly used in Mexico in the future.
Author Response
We appreciate your observations. We have improved the manuscript accordingly.
Round 2
Reviewer 1 Report
Comments and Suggestions for Authors
Major concerns 1 and 2 are not fully addressed, and the Figures of histopathological lesions should be provided. Even though this manuscript has been extensively revised, the current version cannot be recommended for publication.
Author Response
We are thankful for the feedback provided. The response is given below.
Regading major concern 1, our reply after the first revision was that we had changed the title “Immunogenicity and protective efficacy of E. coli-expressed virus-like particles carrying a porcine circovirus type 2b peptide” to “Virus-like particles carrying a porcine circovirus type 2b peptide induce an antibody response and reduce viral load in immunized pigs”, highlighting the most relevant results shown in the manuscript. With respect to major concern 2, our first reply was that it was unlikely, but not impossible, that the adjuvant by itself elicits a specific response towards a PCV2 peptide. With respect to cytokines, we added in the revised manuscript (lines 348-352) that an alternative assay using peripheral blood mononuclear cells re-stimulated with the VLPs and measuring cytokine secretion would have been more appropriate to address the role of the VLP, not the adjuvant, in the cytokine response. We are sorry that we have no other way to address this concern.
As requested, we have included representative micrographs showing the lesions in Supplementary Figure 2.
Once again, we truly appreciate the criticism to improve the manuscript's quality.
Reviewer 3 Report
Comments and Suggestions for Authors
Hernández-Aviña et al, have answered the majority of the queries raised during peer review, however there are some points that have not been amended in the manuscript which would be pertinent for the readership.
Major comments
Whilst PEG8000 does not actively precipitate LPS, LPS has been shown to bind to VLPs, which leads to the co-purification of LPS with VLPs, even following PEG8000 precipitation. The authors should note at the end of their new section describing VLP production and purification that they were not tested for endotoxin levels for full disclosure.
Section 3.2 - Unfortunately I do not think I provided a clear comment on this section originally. The addition of new Figure 1 is very helpful for clarity to the readership, however Section 3.2 still does not provide a clear analysis on the differences in the cytokines between vaccinated and unvaccinated animals pre and post-challenge, only a table of cytokine amounts.
Could the authors provide a more detailed section highlighting the key differences between vaccinated and unvaccinated animals, pre and post-challenge? Such as highlighting key differences and whether such profiles were expected and whether match a Th1 or Th2 response, and whether these readouts may impact the overall efficacy of their vaccine candidate?
Section 3.4 - I appreciated the authors well considered response to my previous query, however the authors have not alluded to this in the manuscript. Could the authors please add this reasoning into the discussion section. This would provide he readership with a clear potential explanation for these differences.
Conclusions - Similarly, I appreciated the response from the authors to my previous comment (quoted below) I would like to see the authors highlight some of these points into their discussion/conclusion sections of the manuscript to provide a commentary on the potential hurdles/requirements to make these VLPs a viable future vaccine candidate.
Previous comment: 'Whilst the authors show the the VLP vaccine reduces viral shedding after challenge, this is also common for numerous vaccines at early time points post-vaccination and this aspect of protection tends to wane over time. Perhaps the authors could reflect on this in their discussion as to how often they would assume the livestock would need to be immunised to provide protection? Additionally, this study did not show that this vaccine reduced disease in these animals, could the authors provide a hypothesis as to whether they think this means this vaccine is unlikely to be effective in the real world, or perhaps suggest some future experiments/trials that may prove the efficacy of this vaccine e.g differing dosing or immunisation site?
Author Response
We really appreciate the constructive criticism provided. Below are the point-by-point responses.
Major comments
Whilst PEG8000 does not actively precipitate LPS, LPS has been shown to bind to VLPs, which leads to the co-purification of LPS with VLPs, even following PEG8000 precipitation. The authors should note at the end of their new section describing VLP production and purification that they were not tested for endotoxin levels for full disclosure.
RESPONSE: We appreciate the clear explanation. We have declared in the manuscript that VLPs were not tested in the VLPs before formulation (line 100).
Section 3.2 - Unfortunately I do not think I provided a clear comment on this section originally. The addition of new Figure 1 is very helpful for clarity to the readership, however Section 3.2 still does not provide a clear analysis on the differences in the cytokines between vaccinated and unvaccinated animals pre and post-challenge, only a table of cytokine amounts.
RESPONSE: We apologize for misunderstanding the request. We have included a description of the cytokine levels found in both non-immunized groups that draw atention (lines 244-248).
Could the authors provide a more detailed section highlighting the key differences between vaccinated and unvaccinated animals, pre and post-challenge? Such as highlighting key differences and whether such profiles were expected and whether match a Th1 or Th2 response, and whether these readouts may impact the overall efficacy of their vaccine candidate?
RESPONSE: We apologize for not having included an explanation in the previous version of the manuscript. We highlighted these differences in lines 341-352 of the last version.
Section 3.4 - I appreciated the authors well considered response to my previous query, however the authors have not alluded to this in the manuscript. Could the authors please add this reasoning into the discussion section. This would provide he readership with a clear potential explanation for these differences.
RESPONSE: We have included this in the manuscript (lines 365-374).
Conclusions - Similarly, I appreciated the response from the authors to my previous comment (quoted below) I would like to see the authors highlight some of these points into their discussion/conclusion sections of the manuscript to provide a commentary on the potential hurdles/requirements to make these VLPs a viable future vaccine candidate.
Previous comment: 'Whilst the authors show the the VLP vaccine reduces viral shedding after challenge, this is also common for numerous vaccines at early time points post-vaccination and this aspect of protection tends to wane over time. Perhaps the authors could reflect on this in their discussion as to how often they would assume the livestock would need to be immunised to provide protection? Additionally, this study did not show that this vaccine reduced disease in these animals, could the authors provide a hypothesis as to whether they think this means this vaccine is unlikely to be effective in the real world, or perhaps suggest some future experiments/trials that may prove the efficacy of this vaccine e.g differing dosing or immunisation site?
RESPONSE: We really thank these suggestions and have highlighted the abovementioned points in the last paragraph of the Discussion Section (lines 375-383).
Round 3
Reviewer 1 Report
Comments and Suggestions for Authors
I am satisfied with the revisions and recommend acceptance of the manuscript in its current form.
Author Response
We appreciate the feedback. It helped significantly to improve the manuscript's quality.